# Determination of Plastic Anisotropy of Extruded 7075 Aluminum Alloy Thick Plate for Simulation of Post-Extrusion Forming

**Dae-Kwan Jung [1,2], Seong-Ho Ha [1], Heung-Kyu Kim [2] and Young-Chul Shin [1,***

[1] Korea Institute of Industrial Technology, Incheon 21999, Korea; loress@kitech.re.kr (D.-K.J.); shha@kitech.re.kr (S.-H.H.)

[2] Department of Automotive Engineering, Kookmin University, Seoul 02707, Korea; krystal@kookmin.ac.kr

*** Correspondence: ycshin@kitech.re.kr; Tel.: +82-32-850-0357

**Abstract:** In this study, the plastic anisotropy distribution of an extruded 7075 aluminum alloy thick plate was evaluated through small-cube compression tests. The extruded plate with a thickness of 15 mm was divided into five layers in order to verify the difference in plastic anisotropy along the thickness direction of the extruded thick plate. Small-cube specimens with a side length of 1 mm were extracted from each layer and subjected to compression tests in each direction to evaluate the directional r-values of the extruded material. The r-values were applied to Hill's quadratic yield criterion to calculate the six coefficients for each layer. To consider the plastic anisotropy in the thickness direction, a finite element model divided into five layers in the thickness direction was applied. Upsetting tests were conducted to verify the accuracy of the finite element analysis using cube specimens with a side length of 15 and 10.6 mm, and the results of the finite element analysis and the upsetting test were compared and analyzed against each other. Consequently, the finite element analyses were precisely simulated the upsetting test results.

**Keywords:** plastic anisotropy; AA7075; FEM; extruded plate; lightweight

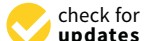



## 1. Introduction

Recently, the conversion of an internal combustion engine vehicle into a new electric one is progressing. However, electric cars use large battery packs that increase body weight. Reducing the weight of parts is necessary for staying competitive in the automobile market. High-strength aluminum alloy has higher specific strengths and appropriate formability when compared with steel. Therefore, it is considered as one of the alternative materials for vehicle weight reduction. In the past, aluminum alloy extrusions were mainly used as building materials. However, as the aluminum alloys with improved strengths to replace steel are being developed, their usage for automotive applications is gradually increasing [1–5]. Aluminum alloy parts are manufactured mainly through sheet and press forming, and bulk forming, such as forging and extrusion. However, the material that undergoes plastic deformation, such as extrusion has strong plastic anisotropy, which varies the mechanical properties depending on the direction and position due to texture changes [6]. Therefore, the plastic anisotropy of extruded products is an important factor in post-extrusion forming [7]. In particular, it has a great effect on the forming process with a large deformation [8]. In order to characterize the anisotropic properties, a many studies have been conducted using numerical models through various functions. Hill [9,10] proposed a quadratic yield criterion with anisotropic coefficients that enables the numerical expression of anisotropy. Later on, many anisotropic yielding functions have been suggested by Hill [11,12], Barlat [13,14], Karafillis [15], and Bassni [16]. However, the aforementioned studies have mainly focused on the examination of rolled sheets. There have been few studies on the plastic anisotropy in the forming of bulk metals such as extrusion [17].

High strength aluminum alloys, such as AA7075, can be potentially used for vehicle components made of steel. AA7075 alloy shows a poor formability that limits its applications in the automotive industry [18–20]. Therefore, more care is needed for process optimization considering the anisotropic features mentioned above. Moreover, since extruded materials have difficulties in evaluating anisotropy through tensile tests due to their dimensional constraints, it is also necessary to consider appropriate mechanical tests. In this study, the deformation behavior of cube-shaped specimens taken from AA7075 alloy extrusions was examined through finite element analysis using Hill's quadratic yield criterion. The Hill's quadratic yield criterion is a function that utilizes the anisotropy coefficient derived through tensile tests. In order to evaluate the three-dimensional anisotropy of extrusion, it is also necessary to take the specimen along the thickness direction, while this can be difficult depending on the thickness of extruded plates. To solve this problem, the yield function calculation using the compression test proposed by Pohlandt et al. [21] was applied. For the determination of localized plastic anisotropy in extrusion, the plastic anisotropy coefficient of extruded material divided into five layers in its thickness direction was derived on the basis of a small-cube compression test proposed by Terano et al. [22,23]. A finite element modeling which reflects the three-dimensional plastic anisotropy of material was carried out by applying the plastic anisotropy coefficient in the form of Hill's quadratic yield criterion to each layer in the thickness of extruded plates, and the accuracy of models was examined by performing a precise FE analysis on the upsetting process of specimens.

## 2. Evaluation of Local Anisotropy for Extruded Plate

### 2.1. Description of Examined Material

The material tested in this study was 7075 aluminum alloy extruded at 450 °C and with an extrusion speed of 1 mm/s and a thick plate of rectangular cross section with a width of 105 mm and a thickness of 15 mm. With regard to the coordinate system of plate, the extrusion direction (ED), transverse direction (TD), and height direction (HD) were defined as X, Y, and Z axes, respectively, as can be seen in Figure 1. According to previous reports [6,24], the 7075 aluminum alloy billet has a random texture distribution, while it shows X-axial symmetric texture after the extrusion.

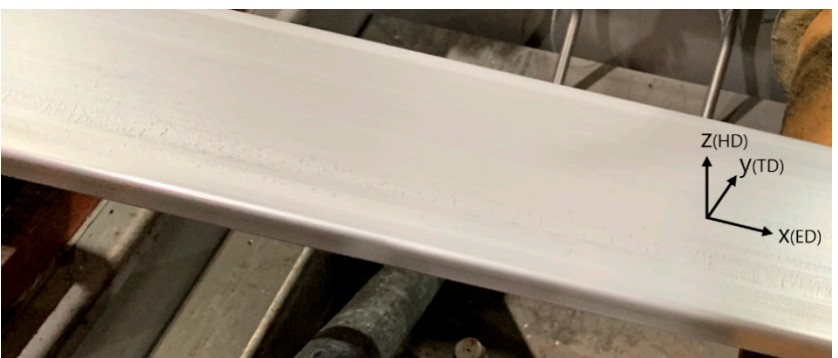

**Figure 1.** Extruded 7075 aluminum alloy thick plate.

### 2.2. Lankford's Value and Hill's Quadratic Yield Criterion

Lankford's value (r-value), which is defined as the ratio of in-plane strain ($\varepsilon_w$) to through thickness strain ($\varepsilon_t$) in the tensile test, is generally used to describe the plastic anisotropy of material.

$$r = \frac{\varepsilon_w}{\varepsilon_t} \tag{1}$$

Since a compression test was performed in this study, the r-value was determined based on the true strain ratio of compressed specimen. For instance, if the extrusion direc-

tion (*X*-axis) is given as $0°$, the r-values are expressed as Equations (2) and (3), assuming that the compression test is performed through $0°$ and $90°$, respectively.

$$r_0 = \frac{d\varepsilon_y}{d\varepsilon_z} = z_0 \tag{2}$$

$$r_{90} = \frac{d\varepsilon_x}{d\varepsilon_z} = z_{90} \tag{3}$$

Hill's quadratic yield criterion was used to describe the anisotropy of the extruded plate. Assuming that the axis of plastic anisotropy is aligned with the X, Y, and Z directions of extruded plate, respectively, the Hill's coefficients can be obtained through simple calculations from the r-values. In addition, the use of Hill's yield criterion allows the numerical expression of plastic anisotropy along the direction and to derive anisotropic constants through compression tests. The Hill's quadratic yield criterion has the following form:

$$2f(\sigma_{ij}) = F(\sigma_y - \sigma_z)^2 + G(\sigma_z - \sigma_x)^2 + H(\sigma_x - \sigma_y)^2 + 2L\tau_{yz^2} + 2M\tau_{zx^2} + 2N\tau_{xy^2} = 2C^2 \tag{4}$$

where *F*, *G*, *H*, *L*, *M*, and *N* are the coefficients characterizing plastic anisotropy. Additionally, *C* is a yield stress in a certain specific direction. *F*, *G*, *H*, *L*, *M*, and *N* are determined from the Equations (5)–(7) on X, Y, and Z planes, respectively.

$$X_\alpha = \left[F + (2L - G - H - 4F)sin^2\alpha cos^2\alpha\right] / \left(Gsin^2\alpha + Hcos^2\alpha\right) \tag{5}$$

$$Y_\beta = \left[G + (2M - H - F - 4G)sin^2\beta cos^2\beta\right] / \left(Hsin^2\beta + Fcos^2\beta\right) \tag{6}$$

$$Z_\gamma = \left[H + (2M - F - G - 4H)sin^2\gamma cos^2\gamma\right] / \left(Fsin^2\gamma + Gcos^2\gamma\right) \tag{7}$$

where $\alpha$, $\beta$, and $\gamma$ are the angles of compression directions. The *F*, *G*, *H*, *L*, *M*, and *N* are calculated by the following Equations (8)–(12).

$$F = X_0 H = H/Z_{90} \tag{8}$$

$$G = Y_{90} H = H/Z_0 \tag{9}$$

$$L = (0.5 + X_{45})(1/X_0 + 1/X_{90})F = (0.5 + X_{45})(1 + 1/Z_0)H, \tag{10}$$

$$M = (0.5 + Y_{45})(1/Y_0 + 1/Y_{90})F = (0.5 + Y_{45})(1 + 1/Z_{90})H \tag{11}$$

$$N = (0.5 + Z_{45})(1/Z_0 + 1/Z_{90})H \tag{12}$$

where *H* is obtained through the Equation (13) derived from the Equations (4), (8) and (9).

$$H = 2Z_0 Z_{90}/(Z_0 + Z_{90}) \tag{13}$$

### 2.3. Small-Cube Compression Test

In this study, it was difficult to fabricate tensile specimens in the thickness direction because the extruded plate examined had a thickness of 15 mm. Therefore, deriving the r-value along the thickness direction was limited. However, the small-cube compression test proposed by Terano et al. [22,23] made it possible to measure the coefficient of plastic anisotropy in the thickness direction. In order to examine the anisotropy change in the thickness direction, the plate was divided into 5 layers in its thickness direction. Considering the symmetry of texture, the cube specimens with a side length of 1 mm were machined in only 3 layers from the center of thickness.

Figure 2 shows the position and orientation of small-cube specimens for the compression test. A total of 27 specimens were taken based on each of the 3 directions from 3 layers. Loading directions for the small-cube compression test in one layer are shown in Figure 3. According to the report by Togh et al. [25], the accurate r-value cannot be obtained at the

compression ratio over 50%. Therefore, the compression ratio for the test was limited to 50%.

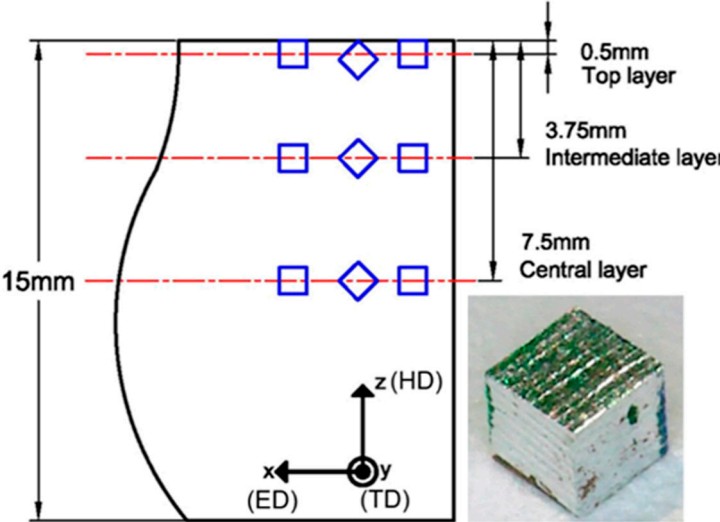

**Figure 2.** Position and orientation of small-cube specimens for compression test.

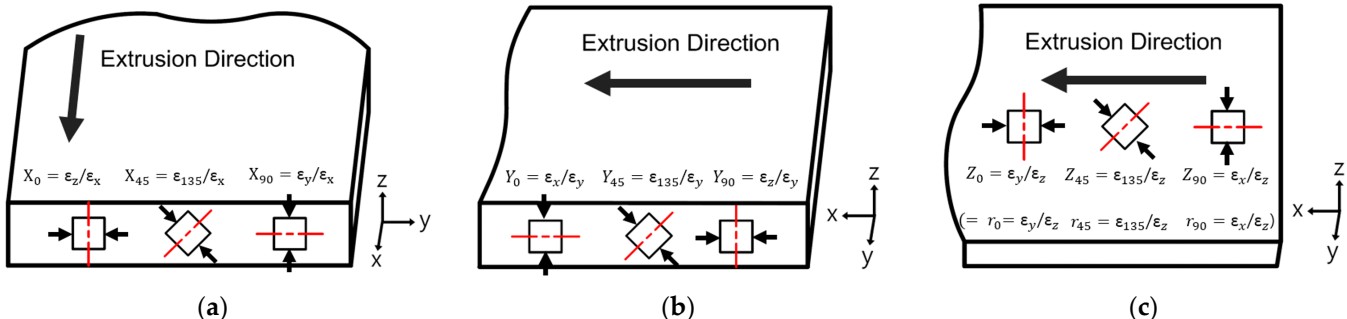

**Figure 3.** Loading directions for small-cube compression test: (**a**) Specimen on X-plane, (**b**) Specimen on Y-plane, (**c**) Specimen on Z-plane.

### 2.4. Results of Small-Cube Compression Test

The deformed shapes of specimens depending on $Y_0$, $Y_{45}$, and $Y_{90}$ directions after the compression test are given in Figure 4. The lengths in each direction for the compressed specimens were measured based on different criteria to evaluate the normal strains. As a result, the measurements from the center of specimens showed remarkable consistencies between experimental results and simulation. Therefore, the normal strains in this study were calculated from the length measurements based on the center of specimens. The r-values depending on layers calculated from the strain length of specimens after the compression test are shown in Table 1. When the plastic anisotropy coefficient is greater or less than 1, it is considered as an anisotropic material. As a result of the test, the r-values were determined as 1.27 in $Y_0$, 1.46 in $Y_{45}$, and 1.54 in $Y_{90}$, respectively. From the values, it is confirmed that the extruded 7075 aluminum alloy plate has a plastic anisotropy. Figure 5 shows the changes in r-values depending on the angles in each layer. Even in the same compression direction, different r-values were obtained depending on the layer, indicating that the anisotropy varies depending on the location.

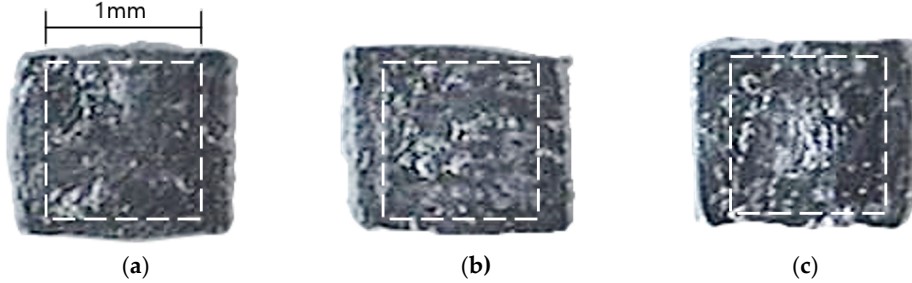

**Figure 4.** Examples of specimens after small-cube compression test: (**a**) $Y_0$, (**b**) $Y_{45}$, (**c**) $Y_{90}$.

**Table 1.** Measured values of compressed small-cube specimens.

| Position | Value | | | | | |
|---|---|---|---|---|---|---|
| | $X_0$ | | $X_{45}$ | | $X_{90}$ | |
| | $\varepsilon_z$ | 0.40 | $\varepsilon_{135}$ | 0.33 | $\varepsilon_y$ | 0.34 |
| | $\varepsilon_x$ | 0.35 | $\varepsilon_x$ | 0.42 | $\varepsilon_x$ | 0.46 |
| | r-value | 1.14 | r-value | 0.79 | r-value | 0.74 |
| | $Y_0$ | | $Y_{45}$ | | $Y_{90}$ | |
| Top layer | $\varepsilon_x$ | 0.46 | $\varepsilon_{135}$ | 0.44 | $\varepsilon_z$ | 0.44 |
| | $\varepsilon_y$ | 0.34 | $\varepsilon_y$ | 0.29 | $\varepsilon_y$ | 0.30 |
| | r-value | 1.35 | r-value | 1.52 | r-value | 1.47 |
| | $Z_0$ | | $Z_{45}$ | | $Z_{90}$ | |
| | $\varepsilon_y$ | 0.30 | $\varepsilon_{135}$ | 0.33 | $\varepsilon_x$ | 0.35 |
| | $\varepsilon_z$ | 0.44 | $\varepsilon_z$ | 0.40 | $\varepsilon_z$ | 0.40 |
| | r-value | 0.68 | r-value | 0.83 | r-value | 0.88 |
| | $X_0$ | | $X_{45}$ | | $X_{90}$ | |
| | $\varepsilon_z$ | 0.40 | $\varepsilon_{135}$ | 0.35 | $\varepsilon_y$ | 0.33 |
| | $\varepsilon_x$ | 0.36 | $\varepsilon_x$ | 0.41 | $\varepsilon_x$ | 0.45 |
| | r-value | 1.11 | r-value | 0.85 | r-value | 0.73 |
| | $Y_0$ | | $Y_{45}$ | | $Y_{90}$ | |
| Intermediate layer | $\varepsilon_x$ | 0.45 | $\varepsilon_{135}$ | 0.42 | $\varepsilon_z$ | 0.46 |
| | $\varepsilon_y$ | 0.33 | $\varepsilon_y$ | 0.29 | $\varepsilon_y$ | 0.31 |
| | r-value | 1.36 | r-value | 1.45 | r-value | 1.48 |
| | $Z_0$ | | $Z_{45}$ | | $Z_{90}$ | |
| | $\varepsilon_y$ | 0.31 | $\varepsilon_{135}$ | 0.32 | $\varepsilon_x$ | 0.36 |
| | $\varepsilon_z$ | 0.46 | $\varepsilon_z$ | 0.43 | $\varepsilon_z$ | 0.40 |
| | r-value | 0.67 | r-value | 0.74 | r-value | 0.90 |
| | $X_0$ | | $X_{45}$ | | $X_{90}$ | |
| | $\varepsilon_z$ | 0.40 | $\varepsilon_{135}$ | 0.34 | $\varepsilon_y$ | 0.33 |
| | $\varepsilon_x$ | 0.39 | $\varepsilon_x$ | 0.41 | $\varepsilon_x$ | 0.42 |
| | r-value | 1.03 | r-value | 0.83 | r-value | 0.79 |
| | $Y_0$ | | $Y_{45}$ | | $Y_{90}$ | |
| Central layer | $\varepsilon_x$ | 0.42 | $\varepsilon_{135}$ | 0.41 | $\varepsilon_z$ | 0.45 |
| | $\varepsilon_y$ | 0.33 | $\varepsilon_y$ | 0.28 | $\varepsilon_y$ | 0.29 |
| | r-value | 1.27 | r-value | 1.46 | r-value | 1.54 |
| | $Z_0$ | | $Z_{45}$ | | $Z_{90}$ | |
| | $\varepsilon_y$ | 0.29 | $\varepsilon_{135}$ | 0.33 | $\varepsilon_x$ | 0.39 |
| | $\varepsilon_z$ | 0.45 | $\varepsilon_z$ | 0.46 | $\varepsilon_z$ | 0.40 |
| | r-value | 0.65 | r-value | 0.72 | r-value | 0.98 |

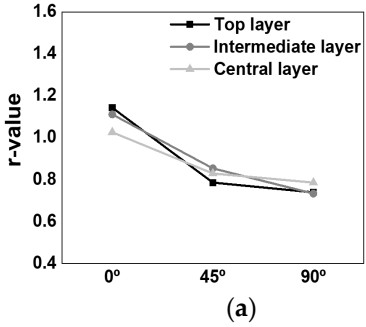 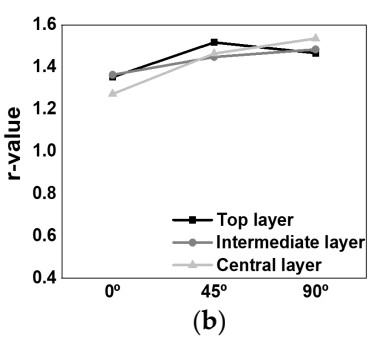 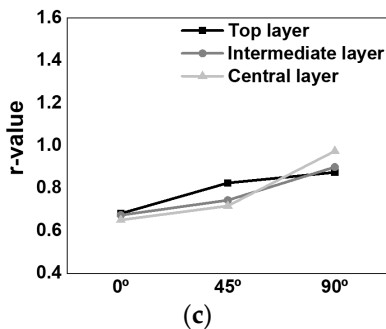

**Figure 5.** Distributions of r-values calculated from each layer: (**a**) Compression on X-plane, (**b**) Compression on Y-plane, (**c**) Compression on Z-plane.

### 2.5. Local Anisotropy for Extruded Plate

By substituting the r-values derived from the compression test into Equations (8)–(13), the Hill's coefficients were calculated. The C value calculated by Equation (4) is given in Table 2.

**Table 2.** Coefficients of Hill's anisotropic yield criterion in three different layers for AA 7075 thick plate.

| Position | F | G | H | L | M | N | C |
|---|---|---|---|---|---|---|---|
| Top layer | 0.88 | 1.28 | 0.77 | 2.43 | 3.31 | 2.65 | $\sigma = 500.8\varepsilon^{0.0582}$ |
| Intermediate layer | 0.86 | 1.27 | 0.77 | 2.59 | 3.17 | 2.49 | $\sigma = 529.8\varepsilon^{0.0652}$ |
| Central layer | 0.80 | 1.23 | 0.78 | 2.63 | 3.11 | 2.43 | $\sigma = 544.9\varepsilon^{0.0648}$ |

## 3. Influence of Plastic Anisotropy on Bulk Forming

Upsetting tests were performed in various directions to investigate the deformation behaviors of extruded thick plates during bulk forming. The specimens used in the test are two types of cubes with side lengths of 15 and 10.6 mm, respectively. The cube specimen with a side length of 15 mm was considered to examine the deformation behavior in the $Z_0$, $Z_{45}$, and $Z_{90}$ directions. The compression ratio was determined as 30% before cracking through a number of tests. In addition, the deformation behavior in the direction rotated by 45° with respect to the x and y axes was examined using the specimen with a side length of 10.6 mm under a compression ratio of 40%. The location and direction where the specimens were taken with the compression direction are shown in Figure 6.

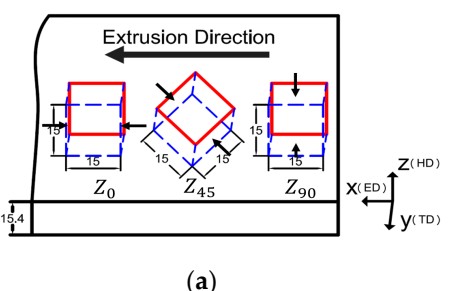 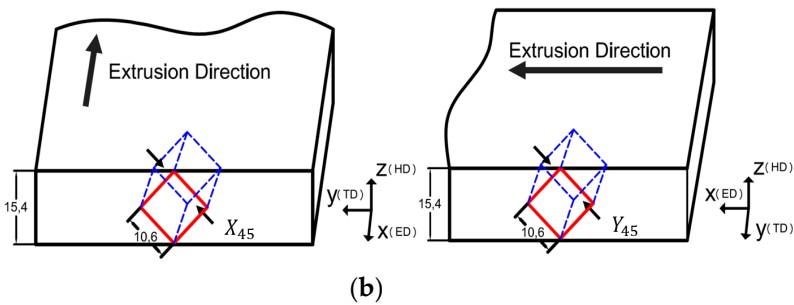

**Figure 6.** Orientations of specimens and loading directions for upsetting test: (**a**) 15 mm cube specimens ($Z_0$, $Z_{45}$, $Z_{90}$), (**b**) 10.6 mm cube specimens ($X_{45}$, $Y_{45}$).

The shapes of specimens after the upsetting tests are shown in Figure 7. The lengths of deformed specimens are given Figure 8, indicating that the deformation behavior varies depending on the compression direction. In the case of the $Z_0$ test, a barreling was observed on the xz plane, while a waistline shape appeared on the xy plane. In the upsetting test along the $Y_{45}$ direction, the asymmetric deformation behavior was found.

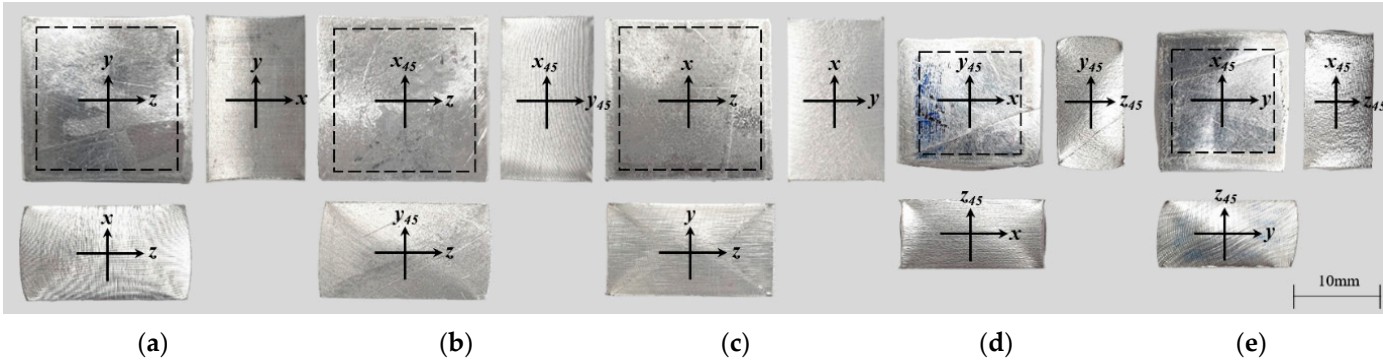

**Figure 7.** Deformed shape of cube specimens after upsetting test: (**a**) $Z_0$, (**b**) $Z_{45}$, (**c**) $Z_{90}$, (**d**) $X_{45}$, (**e**) $Y_{45}$.

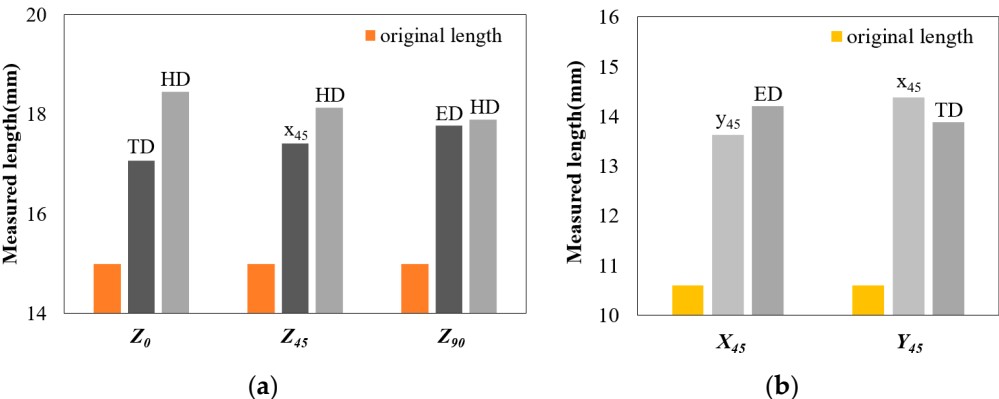

**Figure 8.** Comparisons of measured lengths of compressed cube specimens: (**a**) 15 mm cube, (**b**) 10.6 mm cube.

## 4. Simulation of Thick Plate Upsetting

### 4.1. FE Modeling of Cube Specimens Considering Plastic Anisotropy

Finite element analysis for the upsetting process of AA 7075 extruded plate was conducted using DEFORM-3D, which was developed by SFTC (Columbus, OH, USA). In order to reflect the asymmetry shown in the aforementioned results of the upsetting test, the analysis was carried out using full models, as can be seen in Figure 9. In addition, to apply the anisotropy that varies along the thickness direction, the finite element mode was divided into 5 layers in the thickness direction. The plastic anisotropy values for each layer shown in Table 2 were input. The compression ratio in the upsetting analysis was set to 30% in the 15mm cube and 40% in the 10.6 mm cube, as in the upsetting test. The shear friction coefficient between cube and dies was defined as 0.12.

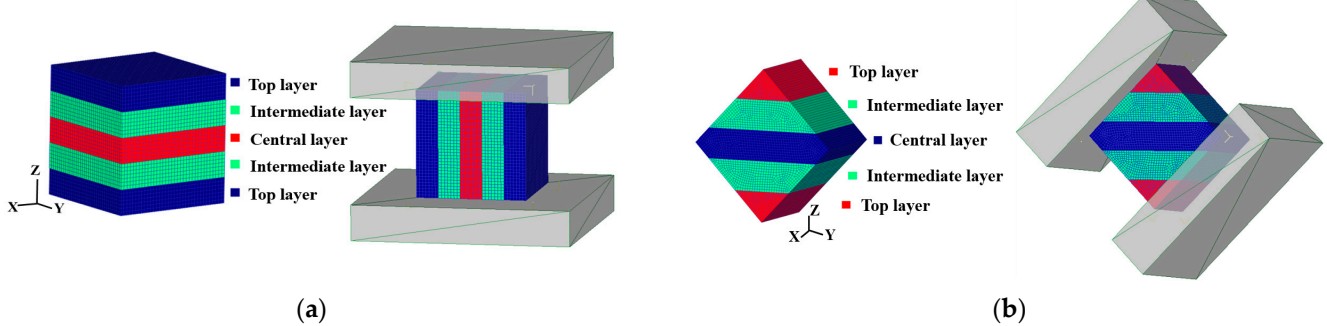

**Figure 9.** FE model of cube specimens considering plastic anisotropy: (**a**) 15 mm cube, (**b**) 10.6 mm cube.

### 4.2. FE Analysis Considering Plastic Anisotropy

The results of the upsetting analysis in the $Z_0$, $Z_{45}$, $Z_{90}$, $x_{45}$, and $y_{45}$ directions are shown in Figure 10. Calculated lengths of compressed cube specimens by FE analysis are given in Figure 11. As a result of the analysis for the 15 mm cube, the y (TD) and z (HD) in $Z_0$ were 17.19 and 18.42 mm, respectively, showing an error of 0.7 to 0.16% from the experimental values. In the case of $Z_{45}$, the $x_{45}$ and $z_{(HD)}$ were 17.47 and 18.14 mm, respectively, with an error of 0.06 to 0.34%. In $Z_{90}$, the $x_{(ED)}$ and $z_{(HD)}$ were 17.72 and 17.89 mm, respectively, showing an error of 0 to 0.28%. Consequently, it was confirmed that the 15 mm cube molding was highly accurate with an error of 0 to 0.7%. In addition, it was found that the 10.6 mm cube analysis was conducted with an error of 0.42 to 2.61%.

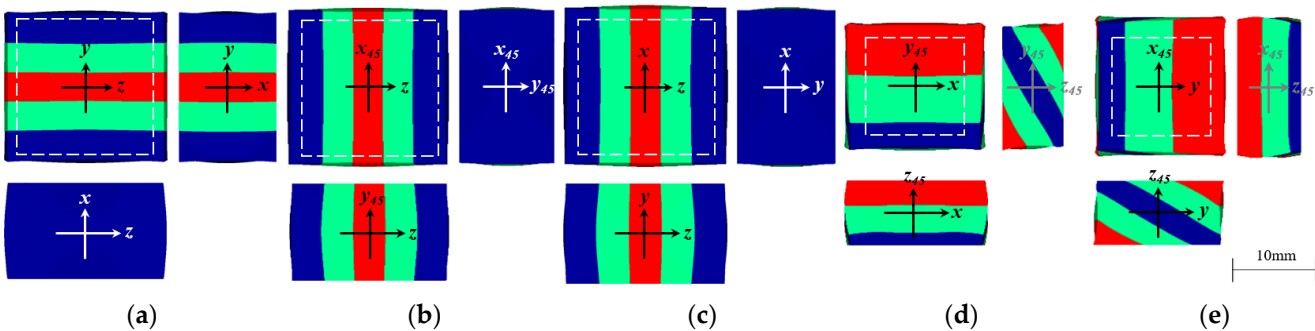

(a)  (b)  (c)  (d)  (e)

**Figure 10.** Results of FE analysis of cube upsetting test: (**a**) $Z_0$, (**b**) $Z_{45}$, (**c**) $Z_{90}$, (**d**) $X_{45}$, (**e**) $Y_{45}$.

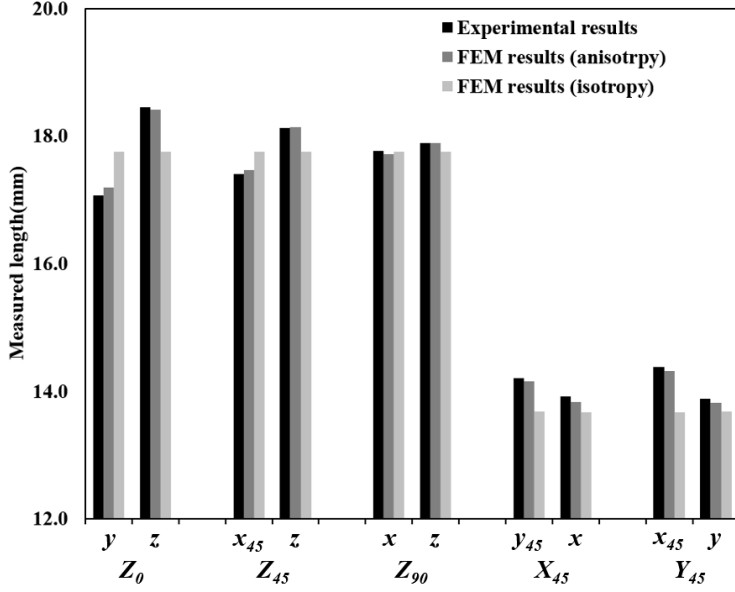

**Figure 11.** Comparison of measured lengths between experimental results and FE analysis.

Figure 11 also shows the experimental results and those by analyses considering isotropy. This indicates that the calculated lengths in the anisotropy analysis are close to those of the experiments, unlike the isotropic analysis.

The shapes of the deformed specimens in the upsetting test and finite element analysis are shown in Figure 12 for comparison. The results of the finite element analysis are in good agreement with those of the experiments, which showed the asymmetric deformation behaviors. Based on the aforementioned results, it is considered that the proposed finite element model is valid for simulating the plastic anisotropy of the extruded 7075 aluminum alloy thick plate.

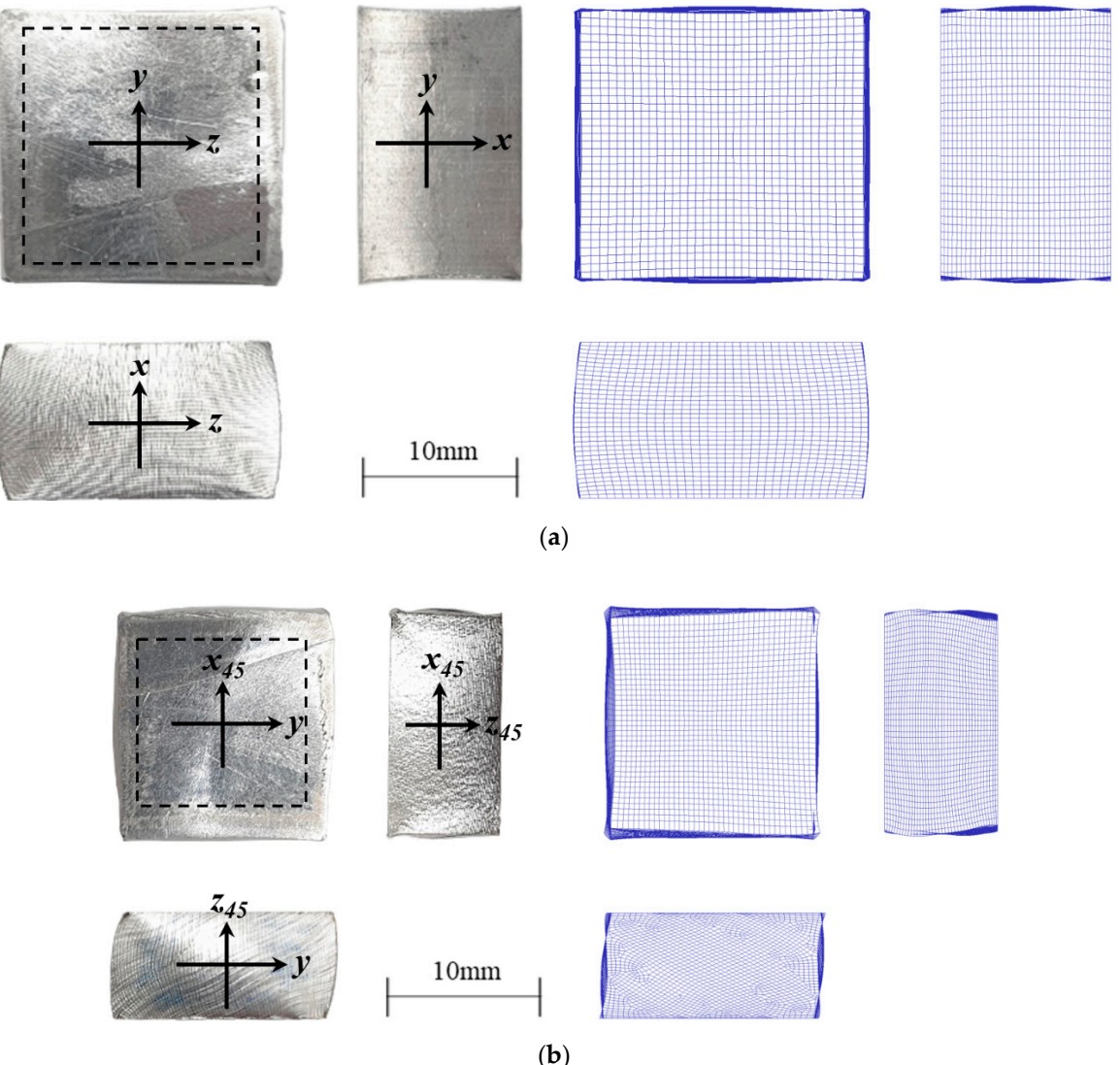

**Figure 12.** Comparison of deformed shapes between experimental results and FE analysis: (**a**) $Z_0$, (**b**) $Y_{45}$.

## 5. Conclusions

In this study, the plastic anisotropy depending on the location and direction of the 7075 aluminum alloy extruded plates was investigated, and the effect of the plastic anisotropy on follow-up plastic working processes was examined through experiments and finite element analysis. The following conclusions were drawn:

1.  It was confirmed that the extruded 7075 aluminum alloy plate had a local plasticity anisotropy not only in the extrusion and transverse directions but also in the thickness direction through the small-cube compression test.
2.  Both the barreling phenomenon and asymmetric deformation behavior were shown in the case of the compression test using the extruded materials. The deformation behaviors of extruded plates were accurately simulated through finite element analysis reflecting Hill's anisotropy coefficients depending on each position and direction.
3.  When considering the production of parts through plastic working, such as forging from the extruded material for weight reduction, the finite element analysis reflecting the local plastic isotropy is necessary for accurate process simulation.

**Author Contributions:** Conceptualization, Y.-C.S.; methodology Y.-C.S. and D.-K.J.; software, D.-K.J.; validation, S.-H.H. and H.-K.K.; formal analysis, D.-K.J.; investigation, D.-K.J.; resources, Y.-C.S. and D.-K.J.; data curation, Y.-C.S. and D.-K.J.; writing—original draft preparation, D.-K.J. and Y.-C.S.;

writing—review and editing, S.-H.H. and H.-K.K.; visualization, D.-K.J.; supervision, Y.-C.S.; project administration, Y.-C.S.; funding acquisition, Y.-C.S. All authors have read and agreed to the published version of the manuscript.

**Funding:** This research was funded by Ministry of Trade, Industry and Energy—grant number 20007282.

**Institutional Review Board Statement:** Not applicable.

**Informed Consent Statement:** Not applicable.

**Data Availability Statement:** Data are available in a publicly accessible repository.

**Conflicts of Interest:** The authors declare no conflict of interest.

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
