# Peer review of "Determination of Plastic Anisotropy of Extruded 7075 Aluminum Alloy Thick Plate for Simulation of Post-Extrusion Forming"

_metals, doi:10.3390/met11040641_

Round 1
Reviewer 1 Report
The authors studied the plastic anisotropy distribution of an extruded 7075 aluminum alloy thick plate. Having read the paper, I am not sure what is the novelty of this work? It should not be the small-cube compression tests which determine the directional r-values. Also, it should not be the application of r-values to Hill's quadratic yield criterion which calculates the 6 coefficients of each layer. All of these are kind of standard tests and known theory. Then FE modelling was conducted and the simulated results (deformed lengths) were compared with the upsetting test, which shows a good agreement. However, still the novelty is not shown. Is it a new modelling method for simulation of anisotropy in the thickness direction the main novelty? If so, the details of modelling method should be given and the difference/novelty should be emphasized. From the title, ‘Determination of plastic anisotropy of extruded 7075 aluminum alloy thick plate for simulation of post-extrusion forming’, it seems the ‘Determination of plastic anisotropy of extruded 7075 aluminum alloy thick plate’ is mainly for simulation of post-deformation which considers the anisotropy, it seems the modelling considering the anisotropy is the main novelty. But as I mentioned it is currently not shown in the text. In the introduction, for applications of aluminium extrusion parts, especially lightweight extrusion profiles/sections with improved strength by novel forming process, I suggest considering a relevant reference (Int. J. Mach. Tools Manuf. 140 (2019), pp.77-88) which could be used to justify ‘as the aluminum alloys with improved strengths to replace steel have been developed, their usage for automotive applications is gradually increasing. Aluminum alloy parts are manufactured mainly through sheet and press forming, and bulk forming such as forging and extrusion.’
Author Response
Point 1: The authors studied the plastic anisotropy distribution of an extruded 7075 aluminum alloy thick plate. Having read the paper, I am not sure what is the novelty of this work? It should not be the small-cube compression tests which determine the directional r-values. Also, it should not be the application of r-values to Hill's quadratic yield criterion which calculates the 6 coefficients of each layer. All of these are kind of standard tests and known theory. Then FE modelling was conducted and the simulated results (deformed lengths) were compared with the upsetting test, which shows a good agreement. However, still the novelty is not shown. Is it a new modelling method for simulation of anisotropy in the thickness direction the main novelty? If so, the details of modelling method should be given and the difference/novelty should be emphasized.
From the title, ‘Determination of plastic anisotropy of extruded 7075 aluminum alloy thick plate for simulation of post-extrusion forming’, it seems the ‘Determination of plastic anisotropy of extruded 7075 aluminum alloy thick plate’ is mainly for simulation of post-deformation which considers the anisotropy, it seems the modelling considering the anisotropy is the main novelty. But as I mentioned it is currently not shown in the text.

Response 1: First of all, authors would like to thank the reviewer for valuable comments and suggestions, which helped us to improve the quality of the paper. As mentioned in ‘Introduction’, the studies for plastic anisotropy of aluminum alloys have mainly focused on the examination of rolled sheets. On the other hand, there have been few studies on the plastic anisotropy in forming of bulk metals such as extrusion. Especially, high strength aluminum alloy AA7075 examined in this study has a poor formability. This article suggests numerical data on the plastic anisotropy of extruded high strength aluminum alloy with a low formability. These points are considered the novelty in this manuscript. More explanations have been made in ‘Introduction’ and highlighted in yellow.
Point 2: In the introduction, for applications of aluminium extrusion parts, especially lightweight extrusion profiles/sections with improved strength by novel forming process, I suggest considering a relevant reference (Int. J. Mach. Tools Manuf. 140 (2019), pp.77-88) which could be used to justify ‘as the aluminum alloys with improved strengths to replace steel have been developed, their usage for automotive applications is gradually increasing. Aluminum alloy parts are manufactured mainly through sheet and press forming, and bulk forming such as forging and extrusion.’
Response 2: As the reviewer suggested, the citation has been added.
Reviewer 2 Report
Paper contains interesting research results but needs a slight improvement:
- The method of presenting the results in Fig. 5 is unclear. In captions (a), (b) and (c), the terms are the same as in the fields of individual graphs - in my opinion, there should be a reference to the X, Y and z directions.
- The comparison of the shape of the samples after swelling, determined experimentally and numerically, is very poorly visible. In my opinion, it would be better to present these differences numerically or on fragments of enlarged sections of 1/4 samples at their corners.
Author Response
Paper contains interesting research results but needs a slight improvement:
Authors would like to thank the reviewer for valuable comments and suggestions, which helped us to improve the quality of the paper.
Point 1: The method of presenting the results in Fig. 5 is unclear. In captions (a), (b) and (c), the terms are the same as in the fields of individual graphs - in my opinion, there should be a reference to the X, Y and z directions.

Response 1: Authors apologize for the serious mistake the reviewer pointed out. The terms are corrected now.
Point 2: The comparison of the shape of the samples after swelling, determined experimentally and numerically, is very poorly visible. In my opinion, it would be better to present these differences numerically or on fragments of enlarged sections of 1/4 samples at their corners.
Response 2: Authors would like to thank the reviewer for the productive comments. The numerical differences are provided in Fig. 11. Authors thought the comparison of 1/4 sections in samples can limit the understanding of overall asymmetric deformation behaviors. Authors have enlarged the images to improve their visibility.
Reviewer 3 Report
This is an interesting paper on the plastic anisotropy of extruded 7075 aluminum alloy and the method is very unique. The followings are comments to improve the quality of the paper.
Figs.4, 7, 12: A scale should be added in each figure.
Edge of specimens are not straight after compression. How did the authors evaluate the length in each direction to calculate the normal strain.
Author Response
This is an interesting paper on the plastic anisotropy of extruded 7075 aluminum alloy and the method is very unique. The followings are comments to improve the quality of the paper:
Authors would like to thank the reviewer for valuable comments and suggestions, which helped us to improve the quality of the paper.
Point 1: Figs.4, 7, 12: A scale should be added in each figure.
Response 1: Authors would like to thank the reviewer for the careful review. The scales have been added in the Figs as reviewer pointed out.
Point 2: Edge of specimens are not straight after compression. How did the authors evaluate the length in each direction to calculate the normal strain.
Response 2: Authors would like to thank the reviewer for the insightful comments. The lengths in each direction for the compressed specimens were measured based on different criteria to evaluate the normal strains. As a result, the measurements from the center of specimens showed remarkable consistencies between experimental results and simulation. Therefore, the normal strains in this study were calculated from the length measurements based on the center of specimens. The details for length evaluation of specimens have been added and highlighted in yellow in the text.
Round 2
Reviewer 1 Report
Thank you for the response, I have no further comments